# Isolation and Expression of Transcription Factors Involved in Somatic Embryo Development by Transcriptome Analysis of Embryogenic Callus of *Thuja koraiensis*

**Chang Ho Ahn** [1,2] , **Jung Yeon Han** [2], **Hyeong Soo Park** [3], **Hyun Won Yoon** [1], **Jung Won Shin** [1] , **Jeong Min Seo** [1], **Hana Lee** [1], **Yeoung Ryul Kim** [1], **Saeng Geul Baek** [1], **Jae Ik Nam** [1], **Jung Min Kim** [1] and **Yong Eui Choi** [2,*]

1    Division of Garden Material Research, Korea Institute of Arboretum Management, Sejong-si 30129, Republic of Korea
2    Department of Forest Resources, College of Forest and Environmental Sciences, Kangwon National University, Chuncheon-si 24341, Republic of Korea
3    Habitat Conservation Division, Korea National Park Service, Gurye-gun 57616, Republic of Korea
\*    Correspondence: yechoi@kangwon.ac.kr

**Abstract:** *Thuja koraiensis* Nakai (Cupressaceae) is an endangered and ecologically important conifer endemic to Korea. Previously, we established a protocol for micropropagation in *T. koraiensis*, which involved somatic embryogenesis from embryogenic callus of *T. koraiensis*. However, the molecular mechanisms underlying somatic embryogenesis remain unclear. Herein, we performed transcriptomic analysis to identify somatic embryogenesis-related genes of *T. koraiensis* via Illumina RNA sequencing. We conducted de novo transcriptome assembly using a Trinity assembler, which produced 274,077 transcript contigs clustered into 205,843 transcripts (unigenes), with an average length of 825 base pairs. Of all the unigenes, 14.69%, 18.62%, and 7.4% had homologs in the Gene Ontology, NCBI Non-redundant Protein, and NCBI Nucleotide databases, respectively. Among these mRNA sequences, expression of putative embryogenesis-associated transcription factors, namely *BABYBOOM* (*BBM*), *WUSCHEL-RELATED HOMEOBOX* (*WOX*), and *SOMATIC EMBRYOGENESIS RECEPTOR-LIKE KINASE* (*SERK*), was analyzed during somatic embryogenesis. RT-PCR analysis revealed that *TkBBM*, *TkWOX*, and *TkSERK* were highly expressed in embryogenic callus and seedling roots, whereas their expression was reduced in seedling leaves. Our findings provide new insights into the roles of *BBM*, *WOX*, and *SERK* in somatic embryogenesis. Our results may serve as a reference for comparative transcriptome analysis in related species and further aid functional genomics studies in conifers.

**Keywords:** Cupressaceae; somatic embryogenesis; Illumina RNA-Seq; de novo assembly; RT-PCR

## 1. Introduction

Gymnosperms first appeared more than 300 million years ago, and currently, approximately 630 conifer species exist on Earth [1]. Conifers are important species of ecological and economic value. Cupressaceae, also known as the cypress family, is distributed throughout the world and comprises >160 species in 32 genera [2]. Many members of the cypress family serve as timber sources and ornamentals and are crucial for the forest ecosystem. Korean arborvitae (*Thuja koraiensis* Nakai), a short and creeping evergreen shrub, which reaches a height of about 3 m, is only found in Korea, predominantly in the high mountains. Recently, *T. koraiensis* was designated rare and endangered due to climate change.

To date, many angiosperm genomes have been sequenced, particularly *Arabidopsis thaliana* [3] and poplar, a woody plant [4]; however, owing to the large size of gymnosperm genomes, reference genomes are still lacking for gymnosperms. Genome assembly has been conducted for only a few members of the cypress family, including *Cunninghamia lanceolate* [5,6], *Juniperus chinensis* [7], *Platycladus orientalis* [8,9], and *T. plicata* [10].

For *T. koraiensis*, Hou et al. [11] performed de novo genome assembly using restriction-associated DNA to develop SSR markers. Nonetheless, no studies have focused on transcriptome analysis in *T. koraiensis*.

Recently, we established a protocol for micropropagation in *T. koraiensis*, which involves somatic embryogenesis (SE) from embryogenic callus [12]. In SE from embryogenic callus, drastic changes take place in the transcriptome, and these changes promote embryogenic potential in plant cells. Among the various factors that are involved in this developmental switch, genes encoding transcription factors (TFs), which constitute sequence-specific DNA-binding proteins and regulate gene expression, are well known to play a central role in the developmental process. Although initiation of SE has been found in many conifers, the molecular mechanisms underlying SE remain unclear. Analyses of gene expression during SE can provide insights into its complex process [13].

TFs, namely *BABYBOOM* (*BBM*) and *WUSCHEL-RELATED HOMEOBOX* (*WOX*), and transmembrane kinase, namely *SOMATIC EMBRYOGENESIS RECEPTOR-LIKE KINASE* (*SERK*), are key regulators of SE initiation and have been identified in both angiosperms and gymnosperms. *BBM* belongs to the APETALA2 (AP2) subfamily of the ETHYLENE-RESPONSE-FACTOR (ERF) TFs [14] and is a key regulator of plant cell totipotency. Moreover, *BBM* has been shown to play a role in developmental pathways associated with cell proliferation and growth [15]. *WOX* forms a large family of genes in plants and has important functions during all stages of plant development [16]. Recently, several members of the *WOX* gene family were reported to be expressed in the early stages of embryogenesis. Haecker et al. [17] and Rupps et al. [18] reported that *WOX* genes were highly expressed during early somatic embryo development in *A. thaliana* and *Larix decidua*, respectively. Similarly, in *Picea abies*, *PaWOX2* expression was demonstrated to be the highest at the earliest stages of somatic embryo development, whereas low levels were observed in seedling tissues [19]. Moreover, *PaWOX2* expression has not been observed in non-embryogenic cell cultures. Therefore, *WOX* genes can serve as potential markers for identifying embryogenic potential. *SERK*, which encodes a leucine-rich repeat containing receptor-like kinase protein, plays a crucial role in the transduction of extracellular signals by phosphorylating intracellular target proteins during SE [20]. *SERK* is expressed in the embryogenic callus of carrots [20] and grapevine [21]. Expression of *SERK* genes in other plant species established a broader expression pattern extending to non-embryogenic, embryogenic, and vegetative tissues [22].

Due to its accuracy and low cost, high-throughput RNA sequencing (RNA-Seq) has been broadly used to analyze transcriptomes both quantitatively and qualitatively. Furthermore, the use of transcriptome sequencing technologies has increased with advancements in genomic technologies. For this reason and to gain insights on the functional genes involved in SE, in the current study, we performed de novo sequencing, assembly, and annotation of the genome sequence of *T. koraiensis* embryogenic callus using Illumina paired-end sequencing technology. Subsequently, on the basis of bioinformatics analysis of assembled transcriptome data, we selected potential embryogenesis-associated genes, namely *BBM*, *WOX*, and *SERK*, and analyzed their functions during SE. Taken together, our findings may serve as a comprehensive sequence resource for other related species and contribute to the understanding of the process of SE in *T. koraiensis*.

## 2. Materials and Methods

### 2.1. Plant Material and Initiation of Embryogenic Culture

In 2018, immature cones were collected from three open-pollinated *T. koraiensis* trees growing in Mt. Seorak (Gangwon province, Korea; at approximately 1590 m; 38°12′23″ latitude, 128°45′79″ longitude) during the northern hemisphere summer. Cones were dissected, immature seeds were extracted, and surfaces of whole seeds were disinfested as follows in the given order: 70% ethanol rinse for 1 min, sterile water rinse for 3 min, 50% Clorox rinse (approximately 4% sodium hypochlorite) for 1 min, and sterile water rinse five times. Seed coats were removed, and whole megagametophytes containing zygotic

embryos were cultured on a semisolid initiation medium (IM) [23], which was modified from an EM medium [24], in Petri dishes (60 × 15 mm). Embryogenic callus lines were maintained by transferring the calluses to fresh IM supplemented with 3% sucrose, 2.25 µM 2,4-dichlorophenoxyacetic acid, and 1.1 µM 6-benzyladenine every 2 weeks.

### 2.2. TruSeq Stranded mRNA and RNA-Seq

For Illumina sequencing, total RNA from embryogenic callus was isolated using Trizol reagent (MRC, Cincinnati, OH, USA) and RNeasy® Plant Mini Kit (QIAGEN, Hilden, Germany). Genomic DNA was removed from the total RNA using DNase I (Takara Bio, Shiga, Japan). Prior to cDNA synthesis, RNA quality and quantity were verified using an Agilent 2100 Bioanalyzer (Agilent Technologies, Santa Clara, CA, USA) and a NanoDrop™ 2000 Spectrophotometer (Thermo Fisher Scientific, Waltham, MA, USA). RNA integrity was confirmed using an Agilent 2100 Bioanalyzer (Agilent Technologies) with a minimum RNA integrity number value ≥ 7. Total RNA was converted into cDNA libraries using the TruSeq Stranded mRNA Sample Prep Kit (Illumina, San Diego, CA, USA) according to the manufacturer's instructions. The cDNA libraries were analyzed for size distribution, quantitated by qPCR (Kapa Library Quantification Kit; Kapa Biosystems, Wilmington, MA, USA), normalized to 2 nmol/L, and then pooled in equimolar amounts for RNA-Seq. Sequencing runs were performed on the HiSeq 4000 (Illumina) according to the method previously described by Bodian et al. [25].

### 2.3. Sequence Trimming, De Novo Transcriptome Assembly, and Annotation

The raw reads were first trimmed to remove the low-quality reads using Trimmomatic software version 0.36 (http://www.usadellab.org/cms/index.php?page=trimmomatic, accessed on 26 June 2018) [26]. The trimmed reads for all samples were pooled and assembled using Trinity software (version r20140717, bowtie 1.1.2), which is generally utilized for the de novo reconstruction of transcriptomes, to obtain high-quality transcript sequences. For gene assembly, the longest contig of the assembled contigs was filtered and clustered into non-redundant transcripts using the CD-HIT-EST program (version 4.6). Subsequently, unigenes were obtained. For the functional annotation of these unigenes, we searched various public databases, including NCBI Nucleotide (NT; http://www.ncbi.nlm.nih.gov/nucleotide/, accessed on 25 June 2018), NCBI Non-redundant Protein (NR; http://www.ncbi.nlm.nih.gov/protein/, accessed on 16 June 2018), Protein Families (Pfam; http://pfam.xfam.org/, accessed on 25 June 2018), Gene Ontology (GO; http://www.geneontology.org/, accessed on 20 June 2018), Universal Protein Resource (UniProt; http://www.uniprot.org/, accessed on 26 June 2018), and EggNOG (http://eggnogdb.embl.de/, accessed on 15 June 2018), using BLASTN of NCBI BLAST (version 2.4.0+) and BLASTX of DIAMOND software (version 0.9.21; http://github.com/bbuchfink/diamond, accessed on 25 June 2018) with an *E*-value default cut-off of $1.0 \times 10^{-5}$. The abundance of the unigenes across samples was estimated using the RSEM algorithm (version v1.2.29, bowtie 1.1.2). The expression levels were calculated as read counts.

### 2.4. Gene Selection and Phylogenetic Analysis of Selected mRNA Sequences

The embryogenesis-associated genes, *BBM*, *WOX*, and *SERK*, were screened using the sequenced transcriptome and NR database. Full-length *BBM*, *WOX*, and *SERK* sequences of other species were obtained from Genebank Non-Redundant database using TBLASTN with TkBBM ("TkBBM1" and "TkBBM2"), TkWOX ("TkWOX1" and "TkWOX2"), and TkSERK ("TkSERK1", "TkSERK2", and "TkSERK3") proteins of *T. koraiensis* as queries. The sequence alignment of *T. koraiensis* proteins and other species (BBM, WOX, and SERK sequences) was compared via phylogenetic tree analysis using the neighbor-joining method with the MEGA v. 7.0.21 (DNA Star, Madison, WI, USA).

*2.5. RT-PCR Analysis*

To assess gene expression in *T. koraiensis* during SE, we collected embryogenic calluses at three different developmental stages and gathered somatic seedling leaves and roots. Total RNA was isolated from the samples using the RNeasy Plant Mini Kit (QIAGEN), and reverse transcription was conducted with the ImProm-II™ Reverse Transcription System (Promega, Madison, WI, USA). As per the RT-PCR, amplification conditions were adjusted as follows: 95 °C for 5 min; 35 cycles of 95 °C for 30 s, 57 °C for 30 s, and 72 °C for 1 min; and a final 7 min extension at 72 °C. The PCR-amplified products were analyzed using 1% (*w/v*) agarose gel electrophoresis, wherein the gel was previously stained with Red-Safe™ DNA dye (iNtRON Biotechnology, Sungnam, Korea) and imaged under GelDoc-ItTS2 Gel Imager (Analytik Jena US LLC, Upland, CA, USA). The intensity of the bands was quantified at least thrice using ImageJ software (National Institutes of Health, Bethesda, MD, USA). To assess RNA integrity, equivalent concentrations of cDNA were utilized; furthermore, actin was used as the internal control. The sequences of all RT-PCR primers are listed in Table S1.

## 3. Results and Discussion

*3.1. RNA-Seq and De Novo Assembly*

A cDNA library constructed from the total RNA extracted from embryogenic cells of *T. koraiensis* was sequenced on an Illumina sequencing platform after validation. To reduce bias in analysis, artifacts, such as low-quality reads, adaptor sequences, contaminant DNA, or PCR duplicates, were removed. Sequencing of *T. koraiensis* produced 250,474,062 reads, which were reduced to 246,686,422 after quality control (Table 1). After trimming to remove low-quality reads, the averages of Q20, Q30, and GC were 98.99, 96.87, and 42.86%, respectively. The de novo transcriptome assembly of *T. koraiensis* performed using the Trinity assembler produced a total of 214,470 transcripts, with an average contig length of 738 nt. Information regarding the number of read bases and the average contig length is shown in Table 2. In total, 205,843 unigenes were obtained, with an average contig length of 619 nt, an N50 of 931 nt, and an N90 of 256 nt. Among the obtained unigenes, 63,599 (30.9%) were ≥500 nt in length, and 29,113 (14.14%) were ≥1000 nt in length. The total length of the assembled sequence was 127,425,530 nt.

**Table 1.** The *T. koraiensis* raw sequenced data and trimmed sequenced data.

| Types of Data | Total Number of Read Bases | Total Number of Reads | GC [a] (%) | Q20 [b] (%) | Q30 [c] (%) |
|---|---|---|---|---|---|
| Raw data statistics | 25,297,880,262 | 250,474,062 | 42.90 | 98.55 | 96.06 |
| Trimming data statistics | 24,768,744,102 | 246,686,422 | 42.86 | 98.99 | 96.87 |

[a] GC content. [b] Ratio of bases that have a Phred quality score greater than or equal to 20. [c] Ratio of bases that have a Phred quality score greater than or equal to 30.

**Table 2.** Summary data of the assembled transcripts and unigenes in *T. koraiensis*.

| Assembly | Transcripts | Unigenes |
|---|---|---|
| Total number of genes | 214,470 | 205,843 |
| GC [a] (%) | 38.79 | 38.35 |
| Contig N90 [b] | 278 | 256 |
| Contig N50 [c] | 1345 | 931 |
| Median contig length | 371.0 | 340.0 |
| Average contig length | 738.06 | 619.04 |
| Total assembled bases | 202,286,040 | 127,425,530 |

[a] GC content. [b] At least 90% of the assembled transcript nucleotides are found in contigs with lengths ≥N90. [c] At least 50% of the assembled transcript nucleotides are found in with lengths ≥N50.

*3.2. Functional Annotation*

The unigenes of the transcriptome were annotated, and sequence similarity analysis was conducted using six databases: NT, NR, Pfam, GO, UniProt, and EggNOG. The greatest

number of annotations were retrieved from the NR database, i.e., 38,320 (18.62%), and the least were retrieved from the NT database, i.e., 15,237 (7.4%) (Table 3). Furthermore, 35,511 (17.25%) unigenes were annotated using EggNOG, and 27,659 (13.44%) unigenes were annotated using Pfam. Additionally, 26,274 (12.76%) unigenes were annotated using the UniProt databases. The size distribution of the unigenes is shown in Figure 1.

**Table 3.** Summary data of the annotation using various databases in *T. koraiensis*.

| Annotation Database | The Number of Annotations | Annotation (%) |
| --- | --- | --- |
| GO [a] | 30,241 | 14.69 |
| UniProt [b] | 26,274 | 12.76 |
| NR [c] | 38,320 | 18.62 |
| Pfam [d] | 27,659 | 13.44 |
| EggNOG | 35,511 | 17.25 |
| NT [e] | 15,237 | 7.40 |
| Overall | 40,461 | 19.66 |

[a] Gene Ontology. [b] Universal Protein Resource. [c] NCBI non-redundant Protein. [d] Protein families. [e] NCBI Nucleotide. The annotations were blasted against public databases with a BLASTN and BLASTX, including NT, Pfam, GO, NR, UniProt, and EggNOG.

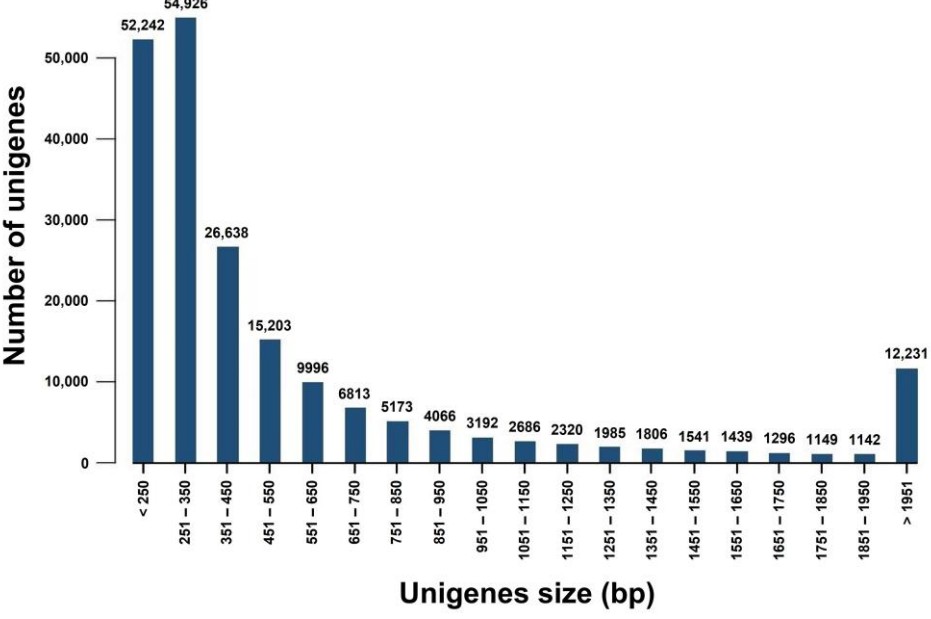

**Figure 1.** Length distribution of all unigenes.

The top-scoring BLASTX hits against the NR protein database showed that 78.2% of the unigenes were present in the genus *Picea*, 11.6% were present in the genus *Pinus*, and 3.9% were present in the genus *Taxus* (Figure 2). Moreover, in the cypress family (Cupressaceae), 1.7% of the unigenes were present in the genus *Cryptomeria*, and 1.3% and 0.7% of the unigenes were present in the genera *Thuja* and *Cunninghamia*, respectively. The most abundant transcript, with 5961 reads, was annotated as a hypothetical protein of *Citrus unshiu*. The 20 most abundant transcripts are listed in Table 4. Two unigenes matched the auxin transport protein in *Amborella trichopoda* and *Quercus suber*.

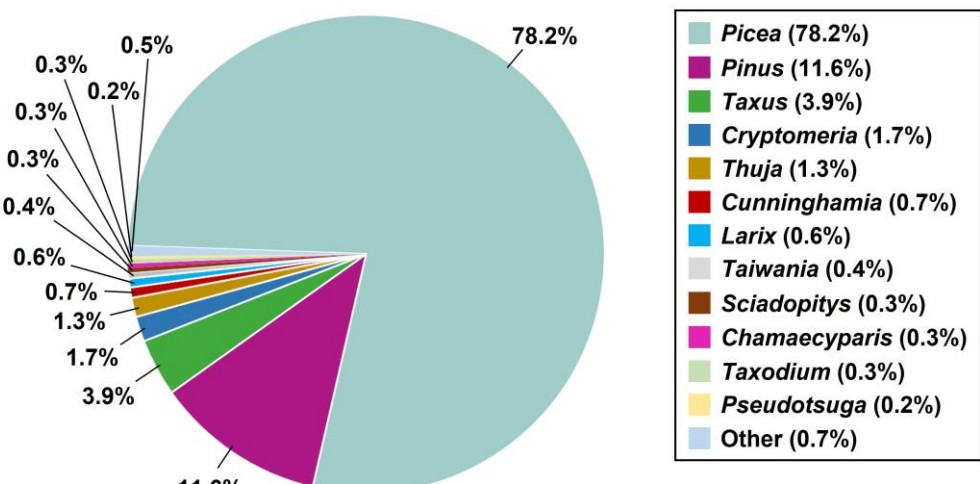

**Figure 2.** Distribution of the top BLASTX hits of conifers for unigenes in the NCBI Non-redundant database.

**Table 4.** Most abundant transcripts in the *T. koraiensis* transcriptome.

| Transcript ID | Read | Target Description (Species) | Identity (%) | *E* Value |
|---|---|---|---|---|
| c121998_g2_i1 | 5961 | Hypothetical protein (*Citrus unshiu*) | 70 | $3 \times 10^{-127}$ |
| c137033_g1_i3 | 5756 | Hypothetical protein (*Marchantia polymorpha*) | 49 | 0 |
| c119709_g1_i3 | 5619 | Hypothetical protein (*Physcomitrella patens*) | 44 | $2 \times 10^{-90}$ |
| c125131_g1_i1 | 5427 | Hypothetical protein (*Plasmodium cynomolgi strain B*) | 24 | 0.0006 |
| c202221_g1_i1 | 5405 | PREDICTED: Midasin (*Ipomoea nil*) | 71 | $3 \times 10^{-10}$ |
| c124649_g1_i2 | 5108 | Auxin transport protein (*Amborella trichopoda*) | 60 | 0 |
| c69094_g1_i1 | 5083 | Auxin transport protein (*Quercus suber*) | 71 | $5 \times 10^{-89}$ |
| c115060_g1_i1 | 4787 | Zinc finger protein (*Macleaya cordata*) | 56 | 0 |
| c121372_g1_i1 | 4506 | PREDICTED: A-kinase anchor protein 9-like isoform X5 (*Musa acuminata* subsp. *malaccensis*) | 24 | $1 \times 10^{-8}$ |
| c139228_g2_i8 | 4360 | Uncharacterized protein (*Amborella trichopoda*) | 50 | 0 |
| c137890_g3_i2 | 4321 | Hypothetical protein (*Marchantia polymorpha*) | 53 | $8 \times 10^{-191}$ |
| c132000_g1_i1 | 4294 | Peroxin/Ferlin domain (*Macleaya cordata*) | 54 | 0 |
| c130259_g1_i3 | 4152 | Hypothetical protein (*Marchantia polymorpha*) | 46 | 0 |
| c138002_g5_i3 | 4128 | Hypothetical protein (*Vitis vinifera*) | 32 | $5 \times 10^{-166}$ |
| c103923_g2_i1 | 4128 | Hypothetical protein (*Vitis vinifera*) | 34 | $7 \times 10^{-21}$ |
| c130469_g2_i4 | 4105 | PREDICTED: Chromatin structure-remodeling complex protein SYD isoform X1 (*Nelumbo nucifera*) | 55 | 0 |
| c134892_g4_i2 | 3977 | Hypothetical protein (*Marchantia polymorpha*) | 75 | 0 |
| c87277_g1_i1 | 3961 | Transformation/transcription domain associated protein (*Trifolium pratense*) | 37 | 0.00008 |
| c134892_g6_i1 | 3896 | PREDICTED: Transformation/transcription domain-associated protein-like (*Nelumbo nucifera*) | 76 | $6 \times 10^{-120}$ |
| c114560_g2_i1 | 3878 | PREDICTED: Uncharacterized protein (*Nicotiana tabacum*) | 33 | $3 \times 10^{-32}$ |

### 3.3. Functional Classification

For functional classification of *T. koraiensis* unigenes, the GO database was employed along with BLASTX of DIAMOND with an E-value cut-off of $1.0 \times 10^{-5}$. The GO terms belonging to biological processes, cellular components, and molecular functions were listed. Subsequently, 30,241 (14.69%) of the total unigenes were assigned GO terms, with 4.68% assignments from biological processes, 5.03% from cellular components, and 4.99% from molecular functions (Figure 3).

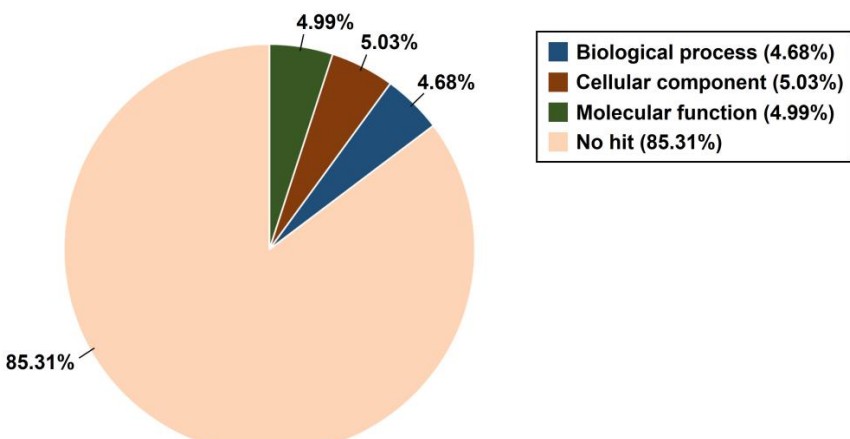

**Figure 3.** Distribution of the Gene Ontology terms for the three categories (biological process, cellular component, and molecular function).

In addition, GO functional classification showed 67 functional groups (Figure 4). The category of biological process comprises a large number of unique sequences that are classified into 35 subcategories (Figure S1). The majority of the sequences in biological processes were involved in metabolic processes (35.61%), followed by those involved in cellular processes (13.09%). The unique sequences were grouped into 18 GO classifications under the cellular component category (Figure S2). Among them, cell part (45.99%) and organelle (22.3%) were the most represented cellular components. The molecular function category included 14 GO classifications (Figure S3). The two largest classifications of molecular function included catalytic activity (48.2%) and binding (33.45%).

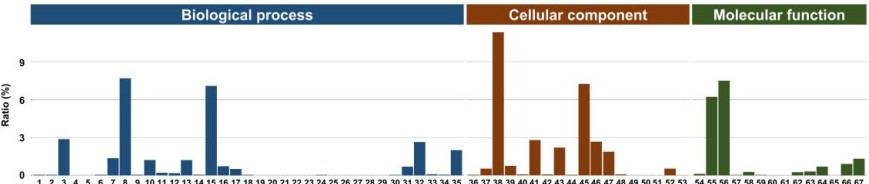

**Figure 4.** Bar diagram of unigenes assigned to 67 Gene Ontology (GO) categories. GO terms of biological process: *1* behavior, *2* biological adhesion, *3* biological regulation, *4* carbon utilization, *5* cell killing, *6* cell proliferation, *7* cellular component organization or biogenesis, *8* cellular process, *9* detoxification, *10* developmental process, *11* growth, *12* immune system process, *13* localization, *14* locomotion, *15* metabolic process, *16* multicellular organismal process, *17* multi-organism process, *18* nitrogen utilization, *19* obsolete chloroplast ribulose bisphosphate carboxylase complex biogenesis, *20* obsolete MAPK import into nucleus, *21* obsolete mitochondrial respiratory chain complex I biogenesis, *22* obsolete mitochondrial respiratory chain complex IV biogenesis, *23* obsolete mycelium development, *24* obsolete protein import into nucleus and docking, *25* obsolete RNA polymerase II complex import into nucleus, *26* obsolete RNA polymerase III complex import into nucleus, *27* obsolete transcription factor import into nucleus, *28* pigmentation, *29* presynaptic process involved in chemical synaptic transmission, *30* reproduction, *31* reproductive process, *32* response to stimulus, *33* rhythmic process, *34* signaling, and *35* unclassified. Cellular component: *36* cells, *37* cell junction, *38* cell part, *39* extracellular region, *40* extracellular region part, *41* membrane, *42* membrane-enclosed lumen, *43* membrane part, *44* nucleoid, *45* organelle, *46* organelle part, *47* protein-containing complex, *48* supramolecular complex, *49* symplast, *50* synapse, *51* synapse part, *52* unclassified, and *53* virion part. Molecular function: *54* antioxidant activity, *55* binding, *56* catalytic activity, *57* molecular carrier activity, *58* molecular function regulator, *59* nutrient reservoir activity, *60* obsolete cyclic pyranopterin monophosphate synthase activity, *61* protein tag, *62* signal transducer activity, *63* structural molecule activity, *64* transcription regulator activity, *65* translation regulator activity, *66* transporter activity, and *67* unclassified.

### 3.4. Identification and Sequence Analysis of TkBBM, TkSERK, and TkWOX

We sought to identify sequences in the transcriptome that encoded authentic *BBM*, *WOX*, and *SERK*. FPKM values of selected contigs for *BBM*, *WOX*, and *SERK* were shown in Figure 5.

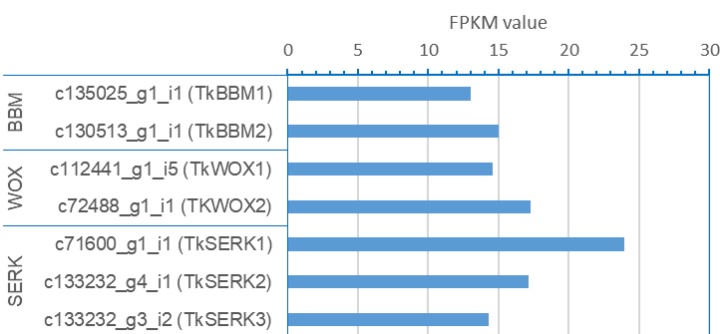

**Figure 5.** FPKM values of selected contigs of *BBM*, *SERK*, and *WOX* in annotated transcriptome sequences of *T. koraiensis* embryogenic callus.

Then, we selected full-length sequences of the coding regions for putative homologs of *BBM*, *WOX,* and *SERK* of *T. koraiensis.* The genes were referred to as *TkBBM* (*TkBBM1* and *TkBBM2*), *TkWOX* (*TkWOX1* and *TkWOX2*), and *TkSERK* (*TkSERK1*, *TkSERK2*, and *TkSERK3*). To examine the phylogenetic similarity between *T. koraiensis* and other plants, we compared the sequences of 57 highly conserved proteins of 28 species.

*BBM* has been reported to be involved in embryogenesis in various plant species, including *A. thaliana* [27], *Brassica napus* [27], *Glycine max* L. [28], *Capsicum annuum* L. [29], *L. decidua* [18], and *Zea mays* [30]. In this study, the open reading frame (ORF) of *TkBBM1* spanned 2325 base pairs (bp) and was predicted to encode a protein of 774 amino acids (aa). The ORF of *TkBBM2* spanned 1791 bp (596 aa). Phylogenetic tree analysis showed the evolutionary relationships between *TkBBM* (*TkBBM1* and *TkBBM2*) and 11 plant species (Figure 6A). *TkBBM* showed a high similarity with *BBM* from other plant species. *TkBBM1* was clearly grouped with *LdBBM* (*L. decidua*, Acc. AEF56566) and *LkBBM* (*Larix gmelinii* var. *olgensis × L. kaempferi aintegumenta*; Acc. AHH34920), whereas *TkBBM2* was observed to be neighboring *RcBBM1* (*Rosacanina*, Acc. AGZ02154). Rupps et al. [18] reported that *LdBBM* showed a similarity of 97.6% to its homolog, *LkBBM*. These findings are similar to our results.

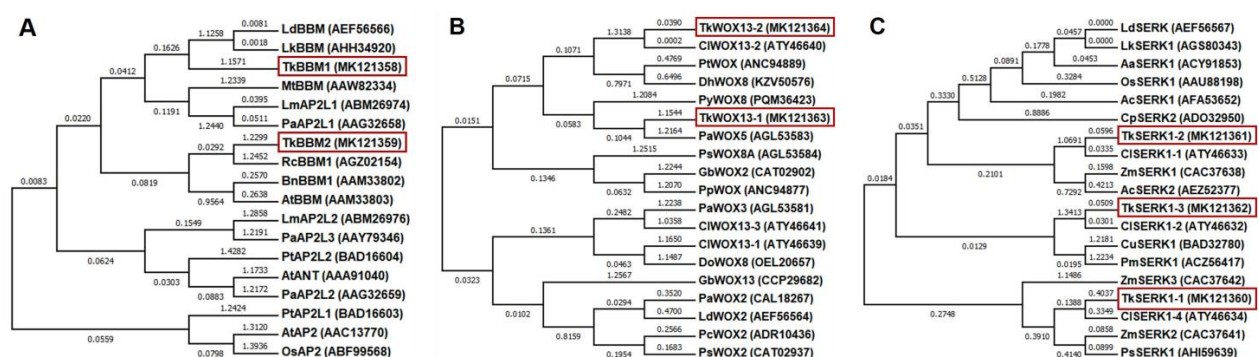

**Figure 6.** Phylogenetic tree of the putative *TkBBM* (*TkBBM1* and *TkBBM2*), *TkWOX* (*TkWOX1* and *TkWOX2*), and *TkSERK* (*TkSERK1*, *TkSERK2,* and *TkSERK3*) proteins based on the NCBI database and representatives characterizing each protein from broad plant lineages. (**A**) *TkBBM1* and *TkBBM2*; (**B**) *TkWOX1* and *TkWOX2*; (**C**) *TkSERK1*, *TkSERK2*, and *TkSERK3*.

Previously, Hedman et al. cloned *WOX* genes in *P. abies* and found that Norway spruce *WOX* genes are represented in all three major clades identified, and that the major diversification within the *WOX* gene family took place before the split between gymnosperms and angiosperms [31]. In this study, the ORF of *TkWOX1* spanned 705 bp and was predicted to encode a 234 aa protein. The ORF of *TkWOX2* spanned 963 bp (320 aa). As per *TkWOX* (*TkWOX1* and *TkWOX2*), eight gymnosperms and three angiosperms were clustered (Figure 6B). *TkWOX1* was closely related to *PaWOX5* (*P. abies*, Acc. AGL53583), and *TkWOX2* was observed to neighbor *ClWOX13-2* (*C. lanceolata*, Acc. ATY46640). According to Van der Graaff et al. [32], the *WOX* gene family was divided into three major clades, namely the modern, intermediate, and ancient clades.

Five genes belonging to the *SERK* family in *A. thaliana* were identified. *SERK1* overexpression has been observed to result in a strong expression of embryogenic competence in transgenic lines [33]. Our results showed that the ORF of *TkSERK1* and *TkSERK2* comprised 1899 bp (632 aa) and 1827 bp (608 aa), respectively, and that the ORF of *TkSERK3* spanned 1845 bp, encoding a 614 aa protein. Expression of *LdSERK* in European larch (*L. decidua*) has been observed to be increased during late embryogenesis [18]. Our results showed that 11 plant species were found to cluster with the *TkSERK* genes (*TkSERK1*, *TkSERK2*, and *TkSERK3*) (Figure 6C). All three *TkSERK* protein sequences showed high similarity to *C. lanceolata* protein sequences (*ClSERK1-1*, Acc. ATY46633; *ClSERK1-2*, Acc. ATY46632; *ClSERK1-4*, ATY46634).

### 3.5. Expression of TkBBM, TkWOX, and TkSERK during SE

For RT-PCR, embryogenic calluses at three different developmental stages and somatic seedling leaves and roots were collected (Figure 7). The expression of *TkBBM*, *TkWOX*, and *TkSERK* during SE was analyzed (Figure 8). In the present study, we observed that the expression of *TkBBM1* and *TkBBM2* were reduced during SE and in somatic seedling leaves. However, both genes showed high expression in somatic seedling roots. These results are in contrast with results reported for *L. decidua*. Rupps et al. [18] reported that the expression of *LdBBM* increased during embryogenesis. Li et al. [34] reported that *LkBBM* showed strong expression in the roots of *L. kaempferi* and *L. olgensis*. In addition, *BBM* groups with PLT1 and PLT2 in the AP2/ERF TF family have been reported to be expressed in the *A. thaliana* root [35]. The *BBM* gene was also identified as an auxin-inducible gene in *Medicago truncatula* roots [36]. Given our results and those of previous studies, we speculate that *TkBBM* plays an important role in regulating the development and growth of roots in *Thuja* spp.

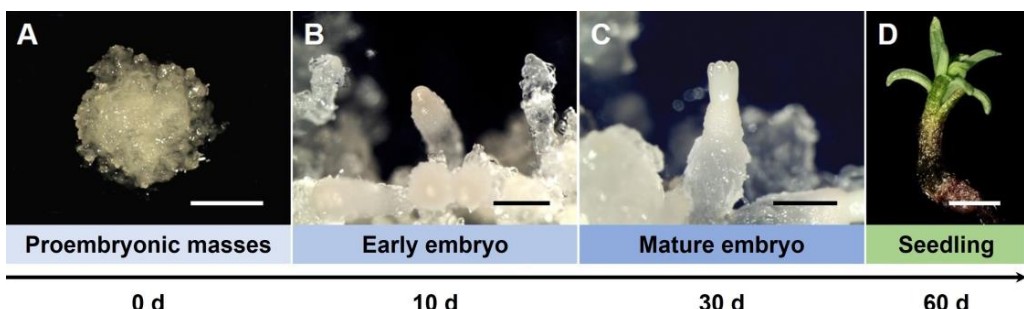

**Figure 7.** Characteristic stages of somatic development of *T. koraiensis* embryos and seedling. (**A**) Proembryonic masses (PEMs) on maturation medium. Scale bar: 5 mm. (**B**) Early embryo on maturation medium after 10 days. Scale bar: 1 mm. (**C**) Fully mature cotyledonary embryo on maturation medium after 30 days. Scale bar: 2 mm. (**D**) Somatic seedling with the first true needles emerging and a taproot on germination medium after 60 days. Scale bar: 1 mm.

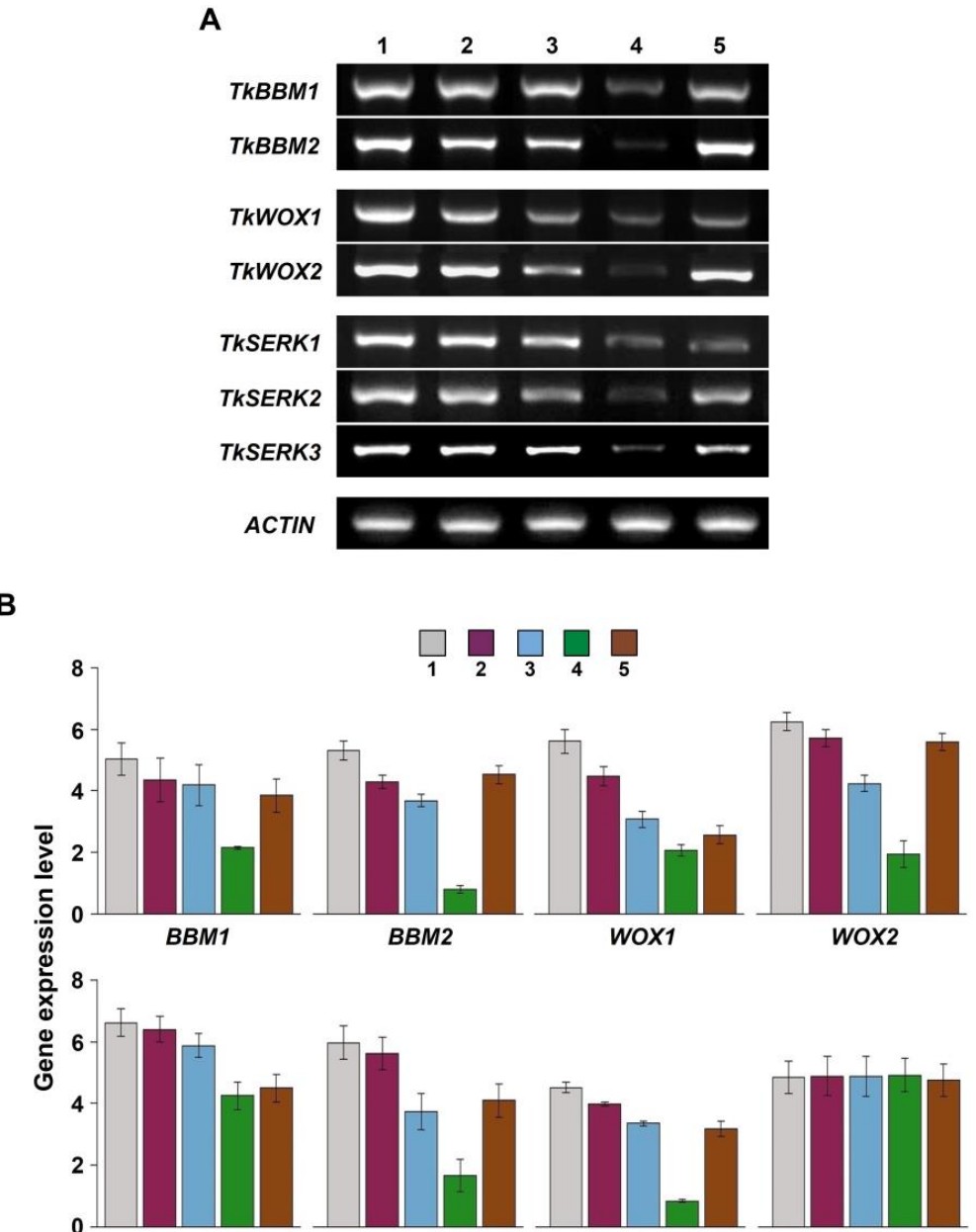

**Figure 8.** Expression analysis of the identified genes during the development of *T. koraiensis* somatic embryos. (**A**) RT-PCR analysis of somatic embryos of different explants. (**B**) Gene expression levels were analyzed, and intensity of bands was quantified by ImageJ densitometry analysis. Gray—proembryonic masses; purple—early embryo; blue—fully mature cotyledonary embryo; green—somatic seedling leaf; and brown—somatic seedling root.

*TkWOX1* and *TkWOX2* were observed to be highly expressed either in embryogenic calluses or at the early stage of somatic embryo development. The expression decreased in parallel with somatic embryo development. *TkWOX1* was similarly expressed in seedling leaves and roots, whereas *TkWOX2* showed faint expression in seedling leaves and strong expression in seedling roots. In the case of *P. abies*, *PaWOX8/9* was observed to be highly expressed in early embryo stages; however, a notably low expression was observed in the mature embryo stage [16]. Similar results were reported for *L. decidua* [18]. *TkSERK1*, *TkSERK2*, and *TkSERK3* were also observed to be highly expressed in early embryogenesis. As compared with the others, *TkSERK1* showed different expression patterns, i.e., uniform expression in seedling leaves and roots. Similarly, *TkSERK2* and *TkSERK3* showed similar

expression patterns. In particular, low expression was detected in seedling leaves. In contrast, *P. glauca SERK1* showed high expression in shoot bud [37]. Only few studies have reported the involvement of *SERK* genes in nodulation.

Owing to the high FPKM values of *TkBBMs*, *TkWOXs*, and *TkSERKs*, as seen in Figure 5, we speculate that *TkBBM*, *TkWOX*, and *TkSERK* are SE-related genes, possibly playing key roles in the SE of *T. koraiensis*. Our study serves as a valuable genetic resource for gene discovery, functional genomics, and comparative genomics in *T. koraiensis*. The *T. koraiensis* sequence resource reported in our study may serve as a reference for comparative transcriptome analysis in related species and aid functional genomics and association studies in conifers. In addition, the findings of the present study enhance our understanding of the transcriptome of *T. koraiensis* during SE.

**Supplementary Materials:** The following supporting information can be downloaded at: https://www.mdpi.com/article/10.3390/horticulturae9020131/s1, Figure S1: Distributions of the GO terms for the biological processes; Figure S2: Distributions of the GO terms for the cellular processes; Figure S3: Distributions of the GO terms for the molecular processes; Table S1: List of primer sequences used for RT-PCR analysis.

**Author Contributions:** Conceptualization and funding acquisition, Y.E.C.; writing—original draft, C.H.A.; formal analysis, J.Y.H.; resources, H.S.P.; methodology, H.W.Y., J.W.S., J.M.S., H.L. and Y.R.K.; software, S.G.B., J.I.N. and J.M.K. All authors have read and agreed to the published version of the manuscript.

**Funding:** This research was funded by the R&D Program for Forest Science Technology provided by the Korea Forest Service (Korea Forestry Promotion Institute), grant number FTIS 2018131B10-1820-BB01 and 2021339A00-2123-CD02.

**Data Availability Statement:** Not applicable.

**Acknowledgments:** We would like to thank the National Park Research Institute of Korea for providing *T. koraiensis* seeds.

**Conflicts of Interest:** The authors declare no conflict of interest.

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
