# Peer review of "Isolation and Expression of Transcription Factors Involved in Somatic Embryo Development by Transcriptome Analysis of Embryogenic Callus of Thuja koraiensis"

_horticulturae, doi:10.3390/horticulturae9020131_

Round 1
Reviewer 1 Report
1. The authors should figure out the concept of 'real-time PCR' firstly.
2. The weight of the manuscript is too low compare to the other similar studies.
3. Logically, the title is not matched with the contents
Author Response
Point 1: The authors should figure out the concept of 'real-time PCR' firstly.
Response 1: Thank you for your kind review. We changed ‘real-time PCR’ to ‘RT-PCR’ in manuscript.
Point 2: The weight of the manuscript is too low compare to the other similar studies.
Response 2: Plants of the genus Thuja are conifers belonging to the cypress family (Cupressaceae) and are distributed in the Americas and Asia. Although there are many other research papers on transcriptome analysis related to somatic embryogenesis, the value of this study is that transcriptome sequencing was performed for the first time in a Thuja species.
Point 3: Logically, the title is not matched with the contents
Response 3: The title is changed to ‘Isolation and expression of transcription factors involved in somatic embryo development by transcriptome analysis of embryogenic callus of Thuja koraiensis’.
Reviewer 2 Report
Transcriptome analysis of somatic embryogenic cells in Thuja
koraiensis Nakai
Chang Ho Ahn, Jung Yeon Han, Hyeong Soo Park, Hyun Won Yoon, Jung Won Shin, Jeong Min Seo, Hana Lee, Yeoung Ryul Kim, Saeng Geul Baek, Jae Ik Nam, Jung Min Kim, Yong Eui Choi
Evaluation.
The authors performed the transcriptomic study of T. koraiensis during its somatic embryogenesis (SE) process. Although this reviewer agrees that there are practically no similar studies on this species and that the approach to the experimental strategy is well structured, the results are analyzed superficially. Hundreds of published articles are now analyzing the transcriptome obtained during SE induction. Since this is the first study of this species, further study of the transcribed genes should be performed.
The authors did a basic analysis of the transcriptome data, including the GO. This analysis is adequate. However, further analysis regarding their genes of interest was not performed. They only did the assembly to identify the sequences. They could have gotten more information from their data.
Validation of the genes that were studied in particular should be performed. In this part of the work, the expression levels were not analyzed. For this reason, it is not seen what the relationship is in the discussion between this part and that of RT-PCR.
The list of primers should go in the supplementary material, and there is no mention of how they evaluated the efficiency of the primers.
Figure 3. What happens to the 85% of unigenes that could not be assigned a GO term?
Figure 6. The accepted nomenclature should be used for the different developmental stages of zygotic embryos. There is a critical point that is frequently seen in the literature. SE is confused with embryo development. I think it would be essential to analyze the articles by Robert B. Goldberg [1,2] so that the authors separate the data between the process of SE and that of embryo development.
Why a relative expression quantification method was not used other than by the intensity of the bands on the gel? This method is not very reliable.
Figure 7. What was the reference to determine the expression "relative"? That is, concerning what the determination made?
Some specific comments are listed below, and the line number in which they are found.
Line 104. Please change 6-benzylaminopurine by 6-benzyladenine.
Lines 122 – 146; 158. Please provide accession dates to the databases and programs. The methodology does not indicate the companies, places of origin, and catalog numbers for all the reagents, only for some.
Line 147. Subtitle 2.5. There is confusion about terms. The authors did not perform real-time PCR (qPCR), in which quantification is performed; they did RT-PCR, which is reverse transcription PCR and is not quantified. Real-time PCR is not the same as RT-PCR.
English language and style are fine/minor spell check required.
Abstract
Thuja koraiensis Nakai (Cupressaceae) is an endangered and ecologically important conifer endemic to Korea. Previously, we established a protocol for micropropagation in T. koraiensis, which involved somatic embryogenesis from the embryogenic callus of T. koraiensis. However, the molecular mechanisms underlying somatic embryogenesis remain unclear. Herein, we performed transcriptomic analysis to identify somatic embryogenesis-related genes of T. koraiensis via Illumina RNA sequencing. We conducted de novo transcriptome assembly using Trinity assembler, which produced 274,077 transcript contigs, clustered into 205,843 transcripts (unigenes), with an average length of 825 base pairs. Of all the unigenes, 14.69%, 18.62%, and 7.4% had homologs in the Gene Ontology, NCBI Non-redundant protein, and NCBI Nucleotide databases, respectively. From among these mRNA sequences, expression of putative embryogenesis-associated transcription factors, namely BABYBOOM (BBM), WUSCHEL-RELATED HOMEOBOX (WOX), and SOMATIC EMBRYOGENESIS RECEPTOR-LIKE KINASE (SERK) was analyzed during somatic embryogenesis. Real-time PCR analysis revealed that TkBBM, TkWOX, and TkSERK were highly expressed in embryogenic callus and seedling roots, whereas their expression was reduced in seedling leaves. Our findings provide new insights into the roles of BBM, WOX, and SERK in somatic embryogenesis. Our results may serve asbe a reference for comparative transcriptome analysis in related species and aid functional genomics studies in conifers.
1. Goldberg, R.B., Beals, T.P., Sanders, P.M. Anther development: basic principles and practical applications. Plant Cell 1993, 5, 1217-1229, 10.1105/tpc.5.10.1217.
2. Goldberg, R.B., De Paiva, G., Yadegari, R. Plant embryogenesis: zygote to seed. Science 1994, 266, 605-614, 10.1126/science.266.5185.605.

Author Response
Point 1: The authors performed the transcriptomic study of T. koraiensis during its somatic embryogenesis (SE) process. Although this reviewer agrees that there are practically no similar studies on this species and that the approach to the experimental strategy is well structured, the results are analyzed superficially. Hundreds of published articles are now analyzing the transcriptome obtained during SE induction. Since this is the first study of this species, further study of the transcribed genes should be performed.
The authors did a basic analysis of the transcriptome data, including the GO. This analysis is adequate. However, further analysis regarding their genes of interest was not performed. They only did the assembly to identify the sequences. They could have gotten more information from their data.
Validation of the genes that were studied in particular should be performed. In this part of the work, the expression levels were not analyzed. For this reason, it is not seen what the relationship is in the discussion between this part and that of RT-PCR.
The list of primers should go in the supplementary material, and there is no mention of how they evaluated the efficiency of the primers.
Response 1: We added one data for FPKM values of sequence data for TkBBM, TkSERK, and TkWOX genes in Figure 5.
The list of primers (Table 1) moved in the supplementary material (Table S1).
We used the actin gene of T. koraiensis for the normalization of gene expressions.
Point 2: Figure 3. What happens to the 85% of unigenes that could not be assigned a GO term?
Response 2: T. koraiensis is an endangered species. Its distribution area is limited, and its population is sparse. There are sporadic distributions only in mountains of 700–1400 m above sea level. Thuja koraiensisis included in the‘Red List of Species in China’as an endangered species of extremely small population. Moreover, no transcriptome data for T. koraiensis also resulted in low GO annotation.
Point 3: Figure 6. The accepted nomenclature should be used for the different developmental stages of zygotic embryos. There is a critical point that is frequently seen in the literature. SE is confused with embryo development. I think it would be essential to analyze the articles by Robert B. Goldberg [1,2] so that the authors separate the data between the process of SE and that of embryo development.
Why a relative expression quantification method was not used other than by the intensity of the bands on the gel? This method is not very reliable.
Response 3: Firstly, we revised the nomenclature of different developmental stages as following the previous SE study of conifer (Smertenko A and Bozhkov PV (2014) somatic embryogenesis: life and death processes during apical-basal patterning. Journal of Experimental Botany). Secondly, we considered qRT-PCR, but the qRT-PCR could not be performed at that time. Thus, gene expression levels were analyzed and the intensity of bands was quantified by ImageJ densitometry analysis.
Point 4: Figure 7. What was the reference to determine the expression "relative"? That is, concerning what the determination made?
Response 4: We changed ‘Relative mRNA expression level’ to ‘Gene expression level’ in Fig. 7.
Point 5: Line 104. Please change 6-benzylaminopurine by 6-benzyladenine.
Response 5: We changed ‘6-benzylaminopurine’ to ‘6-benzyladenine’ in MS.
Point 6: Lines 122 – 146; 158. Please provide accession dates to the databases and programs. The methodology does not indicate the companies, places of origin, and catalog numbers for all the reagents, only for some.
Response 6: We revised them accroding to MDPI instruction for authors.
Point 7: Line 147. Subtitle 2.5. There is confusion about terms. The authors did not perform real-time PCR (qPCR), in which quantification is performed; they did RT-PCR, which is reverse transcription PCR and is not quantified. Real-time PCR is not the same as RT-PCR.
Response 7: We agree with your opinion. We changed ‘real-time PCR’ to ‘RT-PCR’ in the manuscript.
Point 8: English language and style are fine/minor spell check required.
Response 8: We have got English Language Service for editing this MS.

Reviewer 3 Report
Dear authors,
I reckon this is good research. It is great to have a study into plant tissue culture and somatic embryogenesis using NGS analysis. However, there are still major issues needed to be addressed. And I think that you should have more content in your results and discussion. I will suggest accepting the article after revision. Please find my suggestions below:
1. I don’t have many comments in the introduction part. Just feel curious that why you add the author's name in lines 76 and 101 for your citation.
2. Please state what stage of plant tissue culture and how many biologic replicates are for RNA-seq. I will expect to have RNA-seq at different stages. By comparison of However, you might not have it, but please write it more clearly.
3. In section 2.4, you have selected a few genes which related to somatic embryogenesis. I just wonder if you have cloned the genes in this species and might need to double-confirm the accurate sequence by sanger sequencing method.
4. In section 2.5, you stated that is a real-time PCR method. In my point of view, real-time PCR should monitor the exponential value by the fluorescent dye in every cycle, like SYBR green or Taqman methods. According to your method, I think that it can only call semi-quantitative reverse transcriptase PCR (sqRT-PCR). Normally, for sqRT-PCR, different PCR amplification cycles are applied to quantify the intensity of bands.
5. One of the major issue for me is the relation between the transcriptome analysis and the selected gene for expression analysis. For one complete study, I will expect to see the candidate genes selected from your transcriptome analysis. Would you please provide more solid evidence from transcriptome analysis to state the reason for selecting TkBBM, TkSERK, and TkWOX genes.
I have two major comments which are RT-PCR method and how to select candidate genes for expression analysis. I think if you can address these two questions. The logic of this article will be more complete.
Author Response
Point 1: I reckon this is good research. It is great to have a study into plant tissue culture and somatic embryogenesis using NGS analysis. However, there are still major issues needed to be addressed. And I think that you should have more content in your results and discussion. I will suggest accepting the article after revision. Please find my suggestions bellow:
I don’t have many comments in the introduction part. Just feel curious that why you add the author's name in lines 76 and 101 for your citation.
Response 1: Thank you for your kind review. We deleted the author’s name in lines 76 and 101 according to MDPI instructions for authors.
Point 2: Please state what stage of plant tissue culture and how many biologic replicates are for RNA-seq. I will expect to have RNA-seq at different stages. By comparison of however, you might not have it, but please write it more clearly.
Response 2: We used embryogenic callus for RNA-seq. We did not conduct RNA-seq at different stages because of the cost. We could not get biological replication data for RNA-seq, but the RNA-seq was conducted three times with one sample.
Point 3: In section 2.4, you have selected a few genes which related to somatic embryogenesis. I just wonder if you have cloned the genes in this species and might need to double-confirm the accurate sequence by sanger sequencing method.
Response 3: We agree with your opinion. However, because the read counts of selected genes are more than several thousand, we did not confirm the sequence by sanger sequencing.
Point 4: In section 2.5, you stated that is a real-time PCR method. In my point of view, real-time PCR should monitor the exponential value by the fluorescent dye in every cycle, like SYBR green or Taqman methods. According to your method, I think that it can only call semi-quantitative reverse transcriptase PCR (sqRT-PCR). Normally, for sqRT-PCR, different PCR amplification cycles are applied to quantify the intensity of bands.
Response 4: We agree with your opinion. We changed ‘real-time PCR’ to ‘RT-PCR’.
Point 5: One of the major issue for me is the relation between the transcriptome analysis and the selected gene for expression analysis. For one complete study, I will expect to see the candidate genes selected from your transcriptome analysis. Would you please provide more solid evidence from transcriptome analysis to state the reason for selecting TkBBM, TkSERK, and TkWOX genes.
Response 5: In conifer plants, BBM, WOX, and SERK are Key regulators of somatic embryo initiation in conifer species. Although other TFs such as LEAFY COTYLEDON (LEC), FUSCA3 (FUS3), ABSCISIC ACID INSENSITIVE 3 (ABI3), EMBRYOMAKER (EMK) are involved in somatic embryogenesis, these are not involved in early somatic embryogenesis. Thus we did not include these TF genes for analysis in this experiment.

Round 2
Reviewer 2 Report
Table 1 was moved to the supplementary material, but was not deleted from the original text. The following tables were also not renumbered.
Author Response
Point 1: Table 1 was moved to the supplementary material, but was not deleted from the original text. The following tables were also not renumbered.
Response 1: Thank you for your kind review. We revised it and also changed all table’s numbers.
